# Controlled Text Generation as Continuous Optimization with Multiple Constraints

**Sachin Kumar**♣     **Eric Malmi**◇     **Aliaksei Severyn**◇     **Yulia Tsvetkov**♠

♣Language Technologies Institute, Carnegie Mellon University, Pittsburgh, PA, USA
◇Google Research
♠Paul G. Allen School of Computer Science & Engineering, University of Washington
sachink@cs.cmu.edu, {emalmi, severyn}@google.com, yuliats@cs.washington.edu

## Abstract

As large-scale language model pretraining pushes the state-of-the-art in text generation, recent work has turned to controlling attributes of the text such models generate. While modifying the pretrained models via fine-tuning remains the popular approach, it incurs a significant computational cost and can be infeasible due to lack of appropriate data. As an alternative, we propose MuCoCO—a flexible and modular algorithm for controllable inference from pretrained models. We formulate the decoding process as an optimization problem which allows for multiple attributes we aim to control to be easily incorporated as differentiable constraints to the optimization. By relaxing this discrete optimization to a continuous one, we make use of Lagrangian multipliers and gradient-descent based techniques to generate the desired text. We evaluate our approach on controllable machine translation and style transfer with multiple sentence-level attributes and observe significant improvements over baselines.[1]

## 1 Introduction

Recent advances in language models [11, 51, 52] trained on large-scale web text corpora have led to great improvements in state-of-the-art on many natural language processing (NLP) tasks including the ability to generate increasingly coherent text [3]. However, once such models are trained, they are prone to degeneration [66] and biased, non-factual outputs [16, 45] as it is difficult to control the characteristics or attributes of the generated text without architectural modifications [26, 28, 35] and fine-tuning the models on attribute-specific corpora [29, 7]. This can be even more challenging if multiple attributes are involved as labeled data for each combination of attributes can be difficult to obtain.

We focus on *controlled* text generation where the goal is to decode from a text generation model such that the outputs satisfy certain constraints, which the model was not necessarily trained on. For example, given a dialogue generation model, additionally constraining the generated responses to be polite, although the model was not optimized for politeness during training. Recent works address this problem with left-to-right autoregressive decoding, and modify the vocabulary distribution at every step directly using classifiers or language models trained on attribute specific corpora [72, 40, 39], or indirectly via backpropagating gradients through model activations [8]. While exhibiting high level of attribute control, by design these methods can only work with categorical attributes (typically only one attribute) and condition only on the left context while decoding. Additionally, they often require several heuristics to work and are prone to adversarial outputs [64].

---

[1]The code is available at https://github.com/Sachin19/mucoco

35th Conference on Neural Information Processing Systems (NeurIPS 2021).

To address these concerns, we propose the following decoding algorithm. Given a pretrained language model, we posit decoding from it as an optimization problem. First, we relax this discrete optimization problem to a continuous one by representing each token as a simplex on the target vocabulary [19]. This allows us to use continuous optimization techniques like gradient-descent considering each token distribution as parameters; while keeping the language model's parameters fixed (§2). Second, we represent each target attribute to control as a differentiable function. We formulate controllable decoding as a multi-objective optimization problem, with maximizing the log-probability of the language model as well as target attributes as objectives. To make this optimization feasible via gradient-descent, we repurpose it to a constraint optimization problem and solve the dual using the modified differential method of multipliers [48]. We call the algorithm MUCOCO, for incorporating **mu**ltiple **co**nstraints through **c**ontinuous **o**ptimization.

We validate MUCOCO on three conditional text generation tasks with different types of sentence level constraints: (1) Adding formality and cross-lingual similarity in a machine translation model; (2) Ensuring transfer and content-preservation in a style-transfer model, and finally (3) Incorporating multiple styles and attributes (e.g., formality, sentiment magnitude, writer's age group) in a para-phrasing model. With automatic as well as human evaluations we find that our proposed method outperforms strong baselines.

## 2  MUCOCO: Constrained Decoding as Multi-Objective Optimization

For a given language generation task, let $\mathcal{G}$ model the conditional probability $p(\mathbf{y}|\mathbf{x})$ of the output sequence $\mathbf{y} = y_1, \ldots, y_n$, given the input sequence $\mathbf{x} = x_1, \ldots, x_n$. This model can be parameterized using any differentiable architecture like Transformers [62] or LSTMs [20] and trained with any loss function [12, 30], either autoregressively or non-autoregressively [17]. Traditionally, given an input $\mathbf{x}$, decoding from such a model requires finding the output sequence with the highest probability or the lowest negative log-probability, $\mathbf{y}^* = \arg\min_{\mathbf{y}\in\mathcal{Y}} - \log P(\mathbf{y}|\mathbf{x})$. Here $\mathcal{Y}$ is the set of all possible output sequences. In practice, searching $\mathcal{Y}$ to find the highest probability generation is intractable as the space of possible sequences grows exponentially with sequence length and has also been shown to produce undesirable solutions [60]. In most prior work, simple heuristics like beam search, or sampling have been adopted to find approximate solutions, where the text is generated one token at a time (usually left to right) with the output of step $t$ being fed to the input at step $t + 1$.

In this work, however, given $\mathcal{G}$ and an input sequence $\mathbf{x}$, we are interested in finding an output sequence $\mathbf{y}$ that not only maximizes the output probability but also optimizes multiple objectives defined over $\mathbf{x}$ and $\mathbf{y}$. More formally, we seek to find a $\mathbf{y}$ that minimizes all of the following objectives

$$\mathbf{y}^* = \arg\min_{\mathbf{y}\in\mathcal{Y}}(- \log p(\mathbf{y}|\mathbf{x}), f_1(\mathbf{y}), \ldots, f_u(\mathbf{y}), g_1(\mathbf{x}, \mathbf{y}), \ldots, g_v(\mathbf{x}, \mathbf{y})) \tag{1}$$

Here each $f_i$ is a function defined over the output sequence $\mathbf{y}$, for example, the negative log-probability of an attribute (e.g., formality) classifier we want the output sequence to satisfy. And each $g_j$ is a function defined over both the input and output sequence, for example, semantic similarity between $\mathbf{x}$ and $\mathbf{y}$ [55]. We assume all $f_i$ and $g_j$ are differentiable. This is a multi-objective optimization with several possible solutions.

Since there are many objectives to minimize, a left-to-right decoding strategy like beam search or sampling will simply not work due to several reasons. First, the objectives $f_i$ and $g_j$ are sentence-level and hard to define accurately only on generated left-context [72, 40]. Even if we are able to define them, as we add more objectives this process becomes very computationally expensive. Following prior work [19, 50], we formulate this as a continuous optimization process instead of a standard discrete one, and then use standard algorithms for continuous optimization (like gradient descent) for decoding. We maintain a soft-representation of the sequence $\mathbf{y}$, $\tilde{\mathbf{y}} = (\tilde{y}_1, \ldots, \tilde{y}_n)$, where each $\tilde{y}_k \in \Delta_V$ is a simplex over the target vocabulary of size $V$, representing the probability of the $k$-th token. To decode a sentence, we initialize each $\tilde{y}_i$ uniformly over $V$, and treat the entire output sentence as the parameters for gradient descent keeping the parameters of $\mathcal{G}, f_i, g_j$ fixed. After gradient descent has converged, we generate discrete text by selecting the token with the highest probability in $\tilde{y}_k$. We provide more details on the optimization procedure in §2.2.

To make optimization feasible, a multi-objective problem generally yields itself to the following formulation:

$$\arg\min_{y} -\alpha \log p(\mathbf{y}|\mathbf{x}) + \sum_{i=1}^{u} \lambda_i f_i(\mathbf{y}) + \sum_{j=1}^{v} \mu_j g_j(\mathbf{x}, \mathbf{y}), \tag{2}$$

for some statically or dynamically computed weights $\lambda_i$ and $\mu_j$ for each $i$ and $j$, where $\alpha + \sum_i \lambda_i + \sum_j \mu_j = 1$. Although this weighted summation formulation is intuitively appealing, it typically requires an expensive grid-search over the various scalings or use of a heuristic [25, 6, 18]. Furthermore, this formulation by definition assumes a trade-off between the different objectives by essentially assigning an importance weight to each of them. This problem is further exacerbated when different objectives have widely varying scales[2] with smaller scale objectives just getting ignored. More concretely, a multi-objective formulation as we define in (1) admits several possible "optimal" solutions also known as the Pareto set [9]. The image of the Pareto set is called the Pareto front. Since we define all objectives using neural networks, the Pareto front in our case is non-convex, where linear combinations of objectives are shown to be unsuccessful in finding good solutions [36, 37, 10] (see figure 1 for an example).

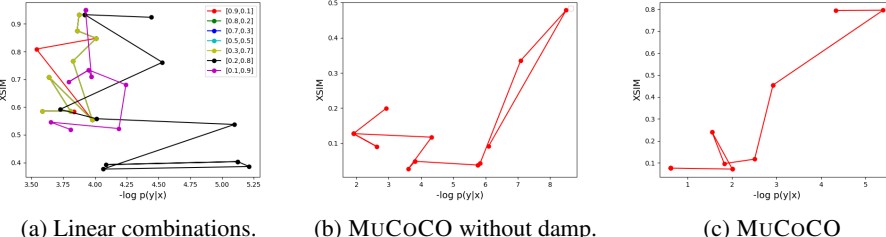

| (a) Linear combinations. | (b) MUCOCO without damp. | (c) MUCOCO |

Figure 1: Loss curves for gradient descent for different configurations for an example of machine translation with a cross-lingual semantic similarity constraint (XSIM $< 0.15$). For each experiment, we do 100 steps of gradient descent (for clarity, we plot the loss values for every 10 steps). See §3.2 for detailed results. Left: In all cases one of the objectives is favored while the other fails to decrease. Middle: We observe fluctuations in the two losses. Right: The losses decrease much more smoothly leading to a better minimum.

Ideally, our goal is a tunable optimization algorithm that finds solutions on the Pareto front, i.e., every solution on the Pareto front should have a hyperparameter value for which the optimization algorithm finds that solution. In order to achieve this, we reframe our optimization problem as a Lagrangian optimization problem instead. We choose one of the losses as the primary objective and consider other losses as constraints. The goal is to minimize the primary loss subject to the secondary losses, each below a threshold value. More formally,

$$\arg\min_{\mathbf{y}} -\log P(\mathbf{y}|\mathbf{x}) \text{ subject to}$$
$$f_i(\mathbf{y}) \le \epsilon_i, i \in \{1, \cdots, u\}$$
$$g_j(\mathbf{x}, \mathbf{y}) \le \xi_j, j \in \{1, \cdots, v\}.$$

Here $\epsilon_i$ and $\xi_j$ are tunable hyperparameters whose values' change can result in different solutions on the Pareto front. This formulation leads to an intuitive interpretation of the decoding process that the generated text from the model $\mathcal{G}$ should satisfy the constraints while being as faithful to the primary objective as much as possible.[3] Consequently, the Lagrangian we end up with looks similar to our original total loss linearly combined as in (2) given by

$$\mathcal{L}(y, \lambda_1, \ldots, \lambda_u, \mu_1, \ldots \mu_v) = -\log p(\mathbf{y}|\mathbf{x}) - \sum_{i=1}^{u} \lambda_i(\epsilon_i - f_i(y)) - \sum_{j=1}^{v} \mu_j(\xi_j - g_j(x, y)) \tag{3}$$

---

[2]For example, classifier log-probabilities are in $(0, \inf)$ while sentence similarities usually lie in [0,1].

[3]For example, defining $f_i(\mathbf{y}) = p(a|\mathbf{y})$ as the probability of a desired attribute $a$ in $\mathbf{y}$ leads to a natural threshold of $f_i(\mathbf{y}) > 0.5$. For a well-calibrated $f_i$, an even higher threshold could be used for inducing highly indicative features of $a$ in $\mathbf{y}$.

where $\lambda_i, \mu_j$ are Lagrange multipliers, and an optimal output $\mathbf{y}^*$ can be obtained as $\mathbf{y}^* = \arg\min_{\mathbf{y}} \max_{\lambda_i \geq 0, \mu_i \geq 0} \mathcal{L}(\mathbf{y}, \lambda_i, \mu_i)$. However, the traditional method of solving the dual function to find $\lambda_i, \mu_j$ that matches $\epsilon_i, \xi_j$, respectively, again leads to a linear trade-off between the various objectives. When the Pareto front is non-convex as in our case, with gradient-descent, the constraints can be ignored and we still cannot always find optimal solutions by tuning $\epsilon_i, \xi_j$ [48].

## 2.1 Modified Differential Method of Multipliers

The fundamental issue in both linear combination of objectives and solving the dual is that fixed scalings $\lambda_i$ and $\mu_i$ (manually pre-determined or obtained by solving the dual) do not work well with gradient descent to minimize for $\mathbf{y}$. Following prior work on differential method of multipliers [48], we propose to use a single gradient descent to optimize for both Lagrangian multipliers and $\mathbf{y}$ simultaneously as follows:

$$\mathbf{y}^{(t)} = \mathbf{y}^{(t-1)} - \eta_1 \nabla_{\mathbf{y}} \mathcal{L}, \lambda_i^t = \lambda_i^{t-1} + \eta_2 \nabla_{\lambda_i} \mathcal{L}, \mu_i^t = \mu_i^{t-1} + \eta_2 \nabla_{\mu_i} \mathcal{L}. \tag{4}$$

We follow the gradient of $\mathcal{L}$ downwards for the $\mathbf{y}$ (descent) and upwards for the multipliers (ascent) while making sure that the multipliers remain positive (by setting the multipliers to 0 whenever they become negative). Intuitively, this algorithm works by increasing the value of the multiplier with each gradient step as long as the constraint is violated. But when the constraint is suddenly satisfied and the multiplier is still large, it might take a number of gradient steps before the gradient descent pushes it to 0, thus causing the solution to be pushed further away from the constraint. As soon as the multipliers become 0 (or negative), the constraint is ignored and the process continues. However when the optimization hits the constraint again, this whole cycle repeats, resulting in "oscillations". We introduce a dampening parameter to each of the multipliers to reduce these oscillations (again following Platt and Barr [48]) and update the Lagrangian as follows:

$$\mathcal{L}(\mathbf{y}, \lambda_i, \mu_j) = -\log p(\mathbf{y}|\mathbf{x}) - \sum_{i=1}^{u} (\lambda_i - \zeta_i)(\epsilon_i - f_i(\mathbf{y})) - \sum_{j=1}^{v} (\mu_j - \nu_j)(\xi_j - g_j(\mathbf{x}, \mathbf{y})), \tag{5}$$

where $\zeta_i = d * \texttt{stop-gradient}(\epsilon_i - f_i(y))$, $\nu_j = d * \texttt{stop-gradient}(\xi_j - g_j(x, y))$ and $d$ is a hyperparameter. $d$ does not affect the final $\mathbf{y}$, just how quickly the algorithm converges to it (We use $d = 1$ in all experiments). $\texttt{stop-gradient}(\cdot)$ indicates that the argument is detached from the computational graph and does not contribute to the gradient computation. When a constraint is not satisfied ($\epsilon_i - f_i(\mathbf{y}) < 0$, hence $\zeta_i < 0$), the dampening parameter $\zeta_i$ being negative incurs higher penalty on the violation than when not using any dampening, without actually increasing the value of $\lambda_i$ too much. But when the constraint is satisfied, it helps quickly reduce the value of penalty being incurred on the constraint while the multiplier converges to 0.

## 2.2 Optimization: Exponentiated Gradient Descent

Our goal is to generate a sequence of discrete symbols $\mathbf{y} = y_1, \ldots, y_T$, where $y_k$ is from the target vocabulary. To make continuous optimization like gradient descent feasible, we adopt a soft-relaxation [19] to represent each $y_k$ as a probability simplex, $\tilde{y}_k \in \Delta_V$ (i.e. $0 \leq \tilde{y}_{kl} \leq 1$ and $\sum_{l=1}^{|V|} \tilde{y}_{kl} = 1$). Intuitively, it gives the probability of each token in the vocabulary. To compute the loss $\mathcal{L}$ during forward pass, we first convert $\tilde{y}_k$ to a one-hot vector $\hat{y}_k$ via a straight through estimator [2]. This allows gradients to be applied to $\tilde{y}_k$ during the backward pass. More formally, $\hat{y}_k = \texttt{one-hot}(\arg\max \tilde{y}_k) - \texttt{stop-gradient}(\tilde{y}_k) + \tilde{y}_k$. During the forward pass, the input embedding tables corresponding to $\mathcal{G}$ and each of the constraints' models receive a one-hot vector $\hat{y}_k$ at each step $k$, and the input embedding is computed as a weighted-sum of the embedding weights. But in the backward pass, the gradients are applied to $\tilde{y}_k$.[4]

This relaxation, however, adds another constraint to the objective $\mathcal{L}$ that each parameter $\tilde{y}_k$ should be a simplex. We use exponentiated gradient descent [27, 19] to solve this problem which modifies the gradient-descent update shown in (4) as: $\tilde{y}_k^{(t)} \propto \tilde{y}_k^{(t-1)} \exp(-\eta_1 \nabla_{\tilde{y}_k} \mathcal{L})$. After every descent step, $\tilde{y}_k^{(t)}$ is normalized to make it a simplex.

---

[4]Unlike prior work [19, 50, 59], we do not feed $\tilde{y}_i$ directly to the model as in our early experiments we found that it leads to slow convergence.

### 2.3 Preventing adversarial solutions: Annealing the thresholds

Finally, it is well known that most neural network based models are not robust to noise and in fact gradient-based methods have been used to generate adversarial examples for text classifiers [59]. We find in our early experiments that using these models to define constraints can also lead to such cases where the constraints are rapidly satisfied but the generated sentences are disfluent. To prevent this issue, we introduce an annealing schedule [47] during the gradient descent where we start with relaxed thresholds $\epsilon_i, \xi_j$ such that they are all satisfied and only the primary loss $-\log p(\mathbf{y}|\mathbf{x})$ is active. As the optimization progresses, we gradually decrease the value of the thresholds causing the constraints to get violated resulting in the optimization gradually shifting to updating $\mathbf{y}$ to satisfy them. The exact schedule we use is described in the next section.

The final decoding algorithm we use in all our experiments is described in the Appendix algorithm 1.

## 3 Experimental Setup

We evaluate MUCOCO on the following controlled generation tasks: reinforcing target style in text generated by a style transfer model §3.1 and adding formality to a machine translation model (§3.2). Additionally, we conduct a qualitative analysis of rewriting a product review to adhere to multiple expected attributes like formality, sentiment magnitude, and age group of the author (§4). These tasks include constraints corresponding to both expected attributes in the target sentence (like formality) as well as both source and target sentences (like semantic similarity) with up to 6 constraints per task.

**Implementation Details** For a given sentence length $T$, we initialize each simplex $\tilde{y}_1, \ldots, \tilde{y}_T$ uniformly over the vocabulary. We use exponentiated descent learning rate of $\eta_1 = 50$ for $\mathbf{y}$ and ascent learning rate of $\eta_2 = 2.0$ for the multipliers, and run the optimization for $100$ steps. Given all intermediate solutions $\mathbf{y}^{(t)}$, we choose the one which satisfies the constraints and has the minimum value of the primary objective. For each constraint, we use the following annealing schedule: we start with an initial value and linearly decrease it at step $40$ until it reaches the desired value at step $80$, after which we keep it constant. Additionally, since the length of the target sequence is not known in advance, we first greedily decode from $\mathcal{G}$ till the end-of-sentence token is generated resulting in a sequence of length $L$. We then use our approach for each $T \in \{L-5, \ldots, L+5\}$ and choose the one which (a) satisfies all the constraints and (b) has the minimum value of the primary objective. However, this optimization objective is highly non-convex and may get stuck in a local minimum where constraints are not satisfied. If none or partial constraints are satisfied, we choose the output based on (b).

### 3.1 Style Transfer

We begin with a style-transfer task, a task aiming to faithfully and fluently rewrite a given sentence such that a desired writing style is reflected in the generation. This task has been widely studied [22, 58, 29, among others] and differs from related tasks like sentiment transfer [61, 33, 34] where flipping the sentiment usually comes at the cost of changing meaning.

Style transfer is usually evaluated across three dimensions: (1) does the output sentence conform to the expected style; (2) does the output sentence preserve the input's meaning; and (3) is the generated sentence fluent. Most prior work in style transfer focused on devising training objectives serving as proxy for the desired outcomes, for example, back-translation [49, 33] or paraphrasing [29] for content preservation and language modeling for style and fluency. But depending on training algorithm and available data, there is often an observed trade-off between transfer and content-preservation [49, 33]. To that end, we add the desired attributes via explicit constraints when decoding from an existing style transfer model.

More specifically, we consider the task of informal to formal transfer [53] with the state-of-the-art unsupervised model STRAP from Krishna et al. [29]. This model is trained in an unsupervised fashion by (1) generating a pseudo-parallel corpus by paraphrasing each formal sentence in the training set (which results in a demotion of stylistic attributes), and (2) training an inverse-paraphrase model to translate paraphrases back to the original formal style. At test time, given an informal input sentence $\mathbf{x}$, the model first generates its paraphrase $\mathbf{z}$, then using an inverse-paraphrase model to generate the output $\hat{\mathbf{y}}$. We train this model by fine-tuning GPT2 (345M) [51] with the GYAFC Corpus

(Entertainment/Music domain; around 50K formal sentences) [53] and evaluate it on the provided test set containing 1312 informal sentences. Krishna et al. [29] report best results with greedy decoding. In MUCOCO we modify the decoding algorithm by considering the negative log-probability of $\mathbf{y}$ given $\mathbf{z}$ according to the model as the primary objective, and incorporate the following constraints:

**Formality**: We train a binary classifier $p_{\text{FORMAL}}(\mathbf{y})$ by fine-tuning GPT2 on the same GYAFC training corpus, following default hyperparameter choices provided in HuggingFace [70]. This classifier outputs the formality probability of a sentence $\mathbf{y}$. We add this output as a constraint to the decoder as $-\log(p_{\text{FORMAL}}(\mathbf{y})) < -\log(0.5)$. In other words, the constraint is satisfied if the classifier assigns at least 0.5 probability of the output $\mathbf{y}$ being formal. We initialize the threshold to 10.0 which is later annealed to $-\log(0.5)$.

**Semantic Similarity**: Since the baseline style-transfer model takes as input the paraphrase $\mathbf{z}$ and not the original text $\mathbf{x}$, it is susceptible to losing some of the original content in $\mathbf{x}$ while generating $\mathbf{y}$. To ensure content preservation we incorporate two kinds of objectives: (1) $\text{USIM}(\mathbf{x}, \mathbf{y}) = \text{cosine}(M(x), M(y))$ [55] where $M$ outputs a continuous vector representation of a given sentence. Similarity between $\mathbf{x}$ and $\mathbf{y}$ is measured by cosine similarity of their respective representations. (2) $\text{WMD}(\mathbf{x}, \mathbf{y})$ takes as input bags of word embeddings of the two sentences and computes the Word Mover's Distance between them [32]. This distance is computed by solving a linear program. We adapt the alternating optimization procedure described in [31] to make this loss differentiable through the program. Intuitively, while USIM computes similarity between sentences taking context into account, it can be less robust to certain missing or repeating tokens, whereas WMD measures lexical overlap between input sentences acting as a proxy for coverage. We discuss the two losses in more detail in Appendix D. To compute the thresholds for constrained optimization, we compute the average value of the two functions on the development set in the same corpus. We use $\text{USIM} \leq 0.15$ and $\text{WMD} \leq 0.4$ as the final constraints (with initial threshold values of 2.0 for each).

**Baselines and Evaluation Metrics** We compare MUCOCO with the following baselines:

NO-CONSTRAINTS: We decode directly from the model greedily without any constraints. This replicates the best result reported by Krishna et al. [29]. We do not use continuous optimization to do unconstrained decoding as it has been shown to perform similarly to left-to-right decoding in prior work [19].

FUDGE: Introduced by Yang and Klein [72], this method decodes in an autoregressive manner. It modifies the output vocabulary distribution at every step by interpolating the language model probability with that of a formality classifier. This classifier is trained to predict the probability of entire sentence being formal given only a prefix (we train it similarly to $p_{\text{FORMAL}}(\mathbf{y})$ by fine-tuning GPT2). This method only works with categorical features like formality and is not extensible to constraints like semantic similarity. We decode using the hyperparameters recommended in [72].

To show the benefits of the constrained optimization setup, we show additional comparisons with a linear combination of objectives in Appendix C

Following the baseline model Krishna et al. [29], we evaluate the generated sentences with the following metrics: (a) **fluency** or grammatical wellformedness measured by the accuracy of a RoBERTa-based classifier model [41] trained on CoLA [65], averaged over all outputs, (b) **transfer**: measured by a RoBERTa-based classifier model [41] trained on the GYAFC training corpus, and finally (c) **WSIM** [68], a subword embedding based similarity model trained on a large-scale paraphrase corpus which performs well on STS benchmarks [4] as well. We measure this metric both with respect to the input and the provided references.[5] In addition, we also report USIM.

**Results** The style transfer results are summarized in table 1. If we only incorporate a formality constraint, we observe that compared to FUDGE our method significantly improves transfer accuracy at the expense of content preservation. Adding semantic similarity constraints on the other hand improves both transfer as well as content preservation with the largest gains achieved when all the constraints are considered together. Qualitative analysis shows that MUCOCO's outputs are typically more fluent and have stronger formality signals, but all of the models are prone to propagating errors from the paraphrasing model (see examples in the Appendix table 4).

---

[5]Each input sentence has 4 references, we choose the highest WSIM value to compute the average.

| Method | Constraint | Fluency | Transfer | Content Preservation (w.r.t. input) | | Content Preservation (w.r.t. ref) | |
|--------|-----------|---------|----------|------|------|------|------|
| | | | | WSIM | USIM | WSIM | USIM |
| STRAP | None | 91% | 78% | 0.69 | 0.77 | 0.72 | 0.80 |
| FUDGE | FORMAL(y) | 90% | 85% | 0.71 | 0.77 | 0.73 | 0.81 |
| MUCOCO | FORMAL(y) | 89% | 93% | 0.67 | 0.75 | 0.72 | 0.78 |
| MUCOCO | USIM(x, y) | 92% | 85% | 0.71 | 0.78 | 0.74 | 0.81 |
| MUCOCO | USIM(x, y), WMD(x, y) | 92% | 87% | **0.73** | **0.79** | **0.77** | **0.86** |
| MUCOCO | SIM(x, y), WMD(x, y), FORMAL(y) | **93%** | **92%** | 0.71 | **0.79** | 0.75 | 0.84 |

Table 1: Automatic evaluation of fluency, formality transfer, and content preservation for informal-to-formal style transfer models.

## 3.2 Style-controlled Machine Translation

We now evaluate MUCOCO in the task of formality transfer in machine translation. Given a trained MT model, decoding is often done using beam search and the highest probability beam candidate is chosen as the final output. Prior work has explored adding rule-based or heuristic constraints such as length penalty or coverage [71] to rerank beam candidates, and adding lexical constraints like penalizing n-gram repetitions [21]. In this experiment, we target sentence-level constraints which are otherwise difficult to incorporate in a left-to-right decoding process. Given a trained MT model and the source text $x$, we use negative log-probability of the translation $y$ under the MT model as our primary objective and incorporate the following constraints for decoding in different combinations:

**Cross-lingual Similarity** Similar to USIM, we define $\text{XSIM}(\mathbf{x}, \mathbf{y}) = \cosine(CM(x), CM(y))$, where $CM$ is a multilingual encoder trained by distilling a monolingual model like $M$ described earlier [56]. More details of training are available in the Appendix D. Averaging across the development set, we use $0.2$ as the threshold for the constraint.

**Formality** Unlike style transfer, where the goal is to rewrite text in the desired style, here we seek to generate translations in a desired style directly from an MT model which was not explicitly trained to conform to a specific style. We train a classifier $p_{\text{FORMAL}}(\mathbf{y})$ similarly to one described in previous section by fine-tuning GPT2, but with a different input-embedding table to match the vocabulary of the decoder of the MT model. Again, we use $\log p_{\text{FORMAL}}(\mathbf{y}) > \log(0.5)$ as the constraint.

**Baselines and Evaluation Metrics** We compare MUCOCO with the following two baselines:

BEAMSEARCH: We decode directly from the translation model with a beam search of size 5.

FUDGE [72]: defined similarly as in the style transfer task but trained to match the decoder vocabulary. As mentioned before, FUDGE only works with categorical attributes like formality and is not easily extensible to constraints like cross-lingual similarity. We use the recommended hyperparameters by Yang and Klein [72] for decoding.

In Yang and Klein [72], the authors also compare FUDGE with other baselines such as PPLM [8] and BEAMSEARCH followed by style transfer. They show that FUDGE vastly outperforms these baselines. Hence, we only show comparisons with FUDGE in this work. We evaluate along the following metrics: (a) **BLEU** [46]: a standard metric for evaluating MT, (b) **BERTScore** [75]: an embedding-based metric which is more robust to changes in surface forms of the words than BLEU. (b) **transfer**: the same RoBERTa-based formality classifier as in our style transfer experiments. We also report XSIM, the constraint we use for decoding.

We experiment with French to English translation with a subset of the OpenSubtitles test set [38] containing 1360 sentence pairs.[6] This test set contains informal spoken language for both source and target. For the primary objective, we use the Marian Transformer based French (fr) to English (en) model [24] through Huggingface. We summarize the results of this experiment in table 2 with selected examples in the Appendix table 5.

---

[6]We create this subset by filtering the original test set to contain only sentence pairs for which beam search translations are classified as informal.

| Method | Constraint | BLEU | BertScore | Formality(%) | XSIM |
|--------|-----------|------|-----------|--------------|------|
| BEAMSEARCH | None | 42.1 | 0.932 | 0% | 0.85 |
| MuCoCO | $\text{XSIM}(\mathbf{x}, \mathbf{y})$ | **42.7** | **0.939** | 4% | **0.88** |
| FUDGE | $\text{FORMAL}(\mathbf{y})$ | 39.2 | 0.922 | 6% | 0.83 |
| MuCoCO | $\text{FORMAL}(\mathbf{y})$ | 37.5 | 0.913 | **30%** | 0.83 |
| MuCoCO | $\text{FORMAL}(\mathbf{y}), \text{XSIM}(\mathbf{x}, \mathbf{y})$ | 39.8 | **0.935** | 23% | **0.86** |

Table 2: Results of style-controlled machine translation experiments.

**Results** By just using a cross-lingual similarity metric without modifying the model at all, we observe +0.6 improvement in BLEU score as well as BERTScore. Adding a formality constraint leads to considerable gain in formality of the outputs with a drop in BLEU; using both XSIM and FORMAL helps recover some of the drop. The drop in BLEU is unsurprising: since BLEU is a surface-level metric it naturally penalizes the translations that are rephrased to conform to formality constraints. Indeed, as shown in table 5, adding a formality constraint leads to changes in sentence structure and vocabulary. On the other hand, we see improvements in BERTScore which is an embedding-based metric, more robust to paraphrasing.

To further validate our results, we conduct a human evaluation of the generated translations. We randomly sample 100 source sentences and their translations generated by beam search and MuCoCO with both FORMAL and XSIM constraints. Two annotators (highly proficient in French and English) to rank the translations on faithfulness (is the source meaning reflected in the translation?) and formality. The options are randomized. On the translation pairs where both annotators agree (79 out of 100), the ones generated by our method were favored by annotators 37% percent of the time, while beam search translations were favored only 18% of the time, and 21% translations were equally favored.

## 4 Discussion

**Simultaneously controlling several attributes** One of the main advantages of our proposed approach is its flexibility to introduce any number of constraints (as long as they are differentiable) to the decoding objective. To illustrate this advantage we consider the following problem: given a sentence annotated with following attributes: age group of the author, formality, and sentiment magnitude, rewrite it such that any chosen combination of the attributes are modified while keeping the others fixed and the content preserved [42, 33]. For our primary objective, we use a inverse-paraphrasing model as defined in §3.1 which we train on a corpus of Yelp Reviews[7] [49]. First, we paraphrase each sentence in the corpus as described in Krishna et al. [29] creating a pseudo-parallel corpus (of reviews and their paraphrases) and train $\mathcal{G}$ as an inverse-paraphrase model to translate the paraphrases back to the original reviews. We use USIM and WMD for semantic similarity constraints and three classifiers for (a) age group of the author (binary; $< 30$ years or $> 30$ years); (b) formality of the review (binary: informal or formal); (c) sentiment magnitude (five-class classifier ratings of 1 to 5). Here we focus on sentiment amplification rather than transfer. That is, changing the 4-star rating of an input to 5 (or 2 to 1). Details of the classifiers and the data used are provided in Appendix D.2.[8] Table 6 shows examples of generated sentences with different combinations of attribute values. We do not focus on sentiment transfer in this setting (e.g. changing a 1-star review to 5-star review) because it also changes the meaning of the utterance making semantic similarity and sentiment constraints incompatible with each other where satisfying one violates the other.

**Finding other solutions on the Pareto front** As described in §2, the thresholds $\epsilon, \xi$ are tunable hyperparameters that allow us to find different solutions on the Pareto front. In our experiments so far, based on expected outcomes and how the constraints are defined, we showed results with only one threshold for each constraint. For example, ideally for a well-calibrated text classifier based constraint, this technique should be able to find solutions for any probability as threshold, but most neural-network based classifiers are not well-calibrated and predict the highest probability output as the label, hence a natural threshold for binary-classifiers is a label probability $> 0.5$. In Appendix

---

[7]This corpus is sentence-tokenized and lowercased with 2.2M sentences not labeled for any attributes.

[8]Due to lack of an established benchmark for this task and due to many possible combinations of attributes, we do not report quantitative results.

table 8, we show how the outputs change if we modify this threshold to different values. We observe that in most cases the optimization converges to generate words more commonly associated with formality. On the other hand, semantic similarity between two sentences is even harder to define, is less robust to noise, and varies with writing styles of the input sentences. As shown, increasing this threshold for semantic similarity can lead to repetitions and disfluency.

**Speed and memory requirements** The presented decoding algorithm treats each token in the output sequence $\mathbf{y}$ as a parameter for gradient-descent which involves multiple forward and backward passes through the primary generative model $\mathcal{G}$ as well as attribute models. Given an expected sequence length $L$, it optimizes $L \times V$ parameters which is both memory and time intensive compared to left-to-right decoding. For example, on a single GeForce RTX 2080 Ti (12GB) on which we run all presented experiments, with a batch size of 1, our approach takes approximately 90 minutes on average to decode around 1200 sentences compared to around 20 minutes for FUDGE [72] with a single constraint. For reference, unconstrained beam-search takes 2-5 minutes. Given enough GPU capacity, however, this approach can easily be extended to larger-batches to improve decoding speed. We do not conduct this experiment due to limited available resources. Using 16-bit floating point operations, this can further be improved. Another way of improving memory efficiency would be to optimize not for tokens directly but instead optimize for token embeddings [30]. This formulation also removes the requirement for all the models to share a vocabulary. We plan to investigate this in future work. Finally, given the capability of this approach to incorporate multiple constraints, it can also be used to generate pseudo-parallel data with different attribute combinations which then could be used to train supervised models for attributes for interest resulting in faster models at inference.

## 5 Ethical considerations

Language generation is a growing research area, and state-of-the-art techniques are still not powerful enough to facilitate fine-grained control over generated content. In the current form, large language models have the potential to generate harmful and biased language. For example, language generators are prone to generating toxic [15] and non-factual content [45], especially when used maliciously [63, 64, 74]. Controlled text generation techniques can be used to mitigate many such problematic biases already encoded in large language models [16, 1, 40]. They also have many other positive use-cases, for example, anonymizing personal attributes in written text [54], and even aiding authors in avoiding implicit biases in their writing [44, 14]. However, none of the existing approaches, including ours, can sufficiently address these issues yet.

We also caution that there are additional risks of adversarial applications of controlled text generation research. The same algorithms that help us control for content preservation and mitigate biases can be used maliciously, to generate misinformation, incorporate pernicious biases, target specific individuals to influence public opinion and seed polarization via manipulating the generated content. For example, when style transfer techniques are used in conjunction with users' personal attributes such as gender, they can amplify harmful social biases. We thus opted not to include gender transfer in our experiments.

Nevertheless, these issues should not discourage the scientific exploration that will advance the state-of-the-art in many positive usages of controlled text generation, including in machine translation, question answering, summarization, dialogue, etc. In parallel, future research should focus on developing better defense methods against mis-using these models maliciously, in a way that could cause societal harms [74].

## 6 Related Work

Recent work on controllable text generation can be divided into two categories. The first focuses on directly training attribute-conditional models either through fine-tuning pretrained models with attribute-specific corpora [13, 29, 5] or via training conditional generative networks [33, 76, 49, 73]. More broadly, this includes methods for text style transfer. Unlike MuCoCo, these methods are not easily extensible and require training or fine-tuning a new model to incorporate new attributes. For example, CTRL [26] train a large scale language model (1.6B parameters) from scratch with 55 control codes capable of generating high-quality text but is very expensive to train. The second line of work, in line with MuCoCo, aims to incorporate control in pre-trained models without

retraining them. For example, GEDI [28] trains smaller class-conditional LMs and uses them as discriminators for guided generation. More recently, FUDGE [72] and DEXPERT [40] propose changes to left-to-right decoding in language models by modifying the vocabulary distribution at every step using attribute classifiers and ensemble of language models trained on attribute-specific corpora. Although lightweight, these approaches are, by design, prone to a trade-off between preserving content and enforcing the attributes in the generated text. Also related to this work is NEUROLOGIC DECODING [43] which uses approximate search in the discrete space (similar to beam search) to satisfy predicate logic constraints. Our work is most closely related to Plug and Play Language Models [8] which use gradients from the attribute models to update the prediction. They work by updating the model activations rather than token probabilities which limits their applicability to only unconditional language models. Furthermore, due to their autoregressive nature, these approaches do not guarantee sequence-level control as they only look at the prefix generated up to a certain step. These are also limited to categorical attributes and can not enforce real-valued controls like semantic similarity.

Gradient-descent based optimization to generate text has been explored in prior work for improving machine translation [19], paraphrasing [50] and generating adversarial examples [59]. These methods however rely on linear combinations of various objectives which as we discuss in §2 are not optimal for non-convex neural-network based models. This phenomenon has also been studied in multi-task learning [36, 37, 57] where linear combination of multiple task losses is the most common approach and approaches for multi-objective gradient descent have been proposed. These approaches can also be explored for text generation in the future.

## 7   Conclusion

We present MUCOCO, a decoding algorithm for controlled generation from (conditional) language models that flexibly combines pretrained LMs with any differentiable constraints. With experiments on style transfer and controlled machine translation, and multiple combination of constraints, we show the effectiveness of this approach. In addition to its potential applications in factual rewriting and debiasing text, this work holds promise in making language generation models personalizable and adaptive to different dialects or even individual speakers, since MUCOCO re-uses pre-trained LMs without adaptation and can incorporate constraints (e.g., dialect or user properties) trained on very little data. Future work will explore more sophisticated optimization techniques to improve the computational efficiency of our approach, and gradient-descent based methods for sampling [67] which will allow to sample from the language models with constraints.

## Acknowledgments

This material is based upon work supported by the National Science Foundation under Grants No. IIS2125201 and IIS2040926, and by the Google faculty research award. The views and opinions of authors expressed herein do not necessarily state or reflect those of the United States Government or any agency thereof. We thank Sebastian Gehrmann for detailed, helpful feedback, Kayo Yin and Artidoro Pagnoni for evaluating our model outputs, Biswajit Paria for helpful discussions, and the anonymous reviewers for much appreciated feedback.

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
