## A  Overview of the Method

Figure 2 shows an overview of our proposed approach.

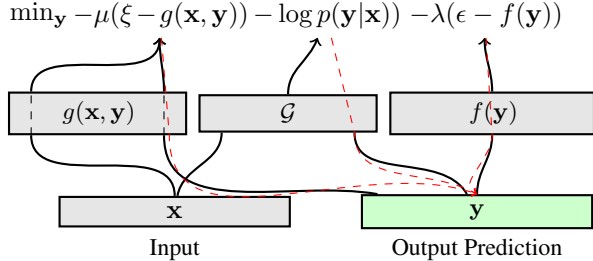

$$\min_{\mathbf{y}} -\mu(\xi - g(\mathbf{x}, \mathbf{y})) - \log p(\mathbf{y}|\mathbf{x}) - \lambda(\epsilon - f(\mathbf{y}))$$

| $g(\mathbf{x}, \mathbf{y})$ | $\mathcal{G}$ | $f(\mathbf{y})$ |

| $\mathbf{x}$ | $\mathbf{y}$ |

Input      Output Prediction

Figure 2: MUCOCO architecture. At each step, only the output sequence $\mathbf{y}$ is updated by receiving gradients from the primary objective of the base text generation model $\mathcal{G}$ as well as the constraints $f$ and $g$, corresponding to arbitrary text attributes to control for at decoding time. Any number of differentiable constraints can be incorporated. Black arrows indicate forward pass while the red dashed arrows indicate the backward pass. The parameters of all the objectives remain frozen (shown in gray).

# B  MUCOCO Decoding Algorithm

---

**Algorithm 1:** MUCOCO: detailed decoding algorithm

---

**Input:** input sequence $\mathbf{x}$, output length $L$, base model $\mathcal{G}$, attribute functions $f_i$ and $g_j$ and their respective initial and final thresholds, threshold update schedule, step sizes $\eta_1, \eta_2$;

**Result:** output sequence $\mathbf{y}$

For all $k \in \{1, \ldots, L\}$, initialize $\tilde{\mathbf{y}}_k^0$ uniformly over $\Delta_V$;

For all $i \in \{1, \ldots u\}$ and $j \in \{1 \ldots v\}$, initialize $\lambda_i^0, \mu_i^0$ as 0 and the thresholds $\epsilon_i^0, \xi_j^0$ with the given values ;

**for** $t = 1, \ldots, \text{MAXSTEPS}$ **do**

    // forward pass

    for all $k$, compute $\hat{y}_k = \texttt{one-hot}(\arg\max \tilde{y}_k)$ and compute the loss $\mathcal{L}$ (using (5));

    // backward pass

    for all $k, i$ and $j$, compute $\nabla_{\tilde{y}_k}^{t-1} = \frac{\partial \mathcal{L}}{\partial \tilde{\mathbf{y}}_k}, \nabla_{\lambda_i}^{t-1} = \frac{\partial \mathcal{L}}{\partial \lambda_i}, \nabla_{\mu_j}^{t-1} = \frac{\partial \mathcal{L}}{\partial \mu_j}$;

    // Update the parameters

    update $\tilde{y}_k^{(t+1)} \propto \tilde{y}_k^{(t)} \exp(1 - \eta_1 \nabla_{\tilde{y}_k} \mathcal{L})$;

    update $\lambda_i^t = \max(0, \lambda_i^{t-1} + \eta_2 \nabla_{\lambda_i} \mathcal{L})$, and $\mu_i^t = \max(0, \mu_i^{t-1} + \eta_2 \nabla_{\mu_i} \mathcal{L})$;

    update $\epsilon_i^t, \xi_j^t$ following the threshold update schedule

**end**

**return** $\arg\min_t \{-\log p(\tilde{\mathbf{y}}^{(t)}|\mathbf{x}) : \forall i, f_i(\tilde{\mathbf{y}}^{(t)}) \leq \epsilon_i, \forall j, g_j(\mathbf{x}, \tilde{\mathbf{y}}^{(t)}) \leq \xi_j\}$;

---

# C  Additional Results

In figure 1b, we gave a motivating example of why linear combination of objectives leads to some of objectives getting ignored. In table 3, for one constraint USIM, we vary the weights of the linear combination and show that to indeed be the case.

# D  Details of Attribute Models

## D.1  Semantic similarity models

We explain the semantic similarity models we use in our experiments in more detail here:

| Weights | | Fluency (%) | Transfer (%) | wsim (w.r.t. input) | wsim (w.r.t. ref.) |
| --- | --- | --- | --- | --- | --- |
| $-\log p(\mathbf{y}|\mathbf{x})$ | USIM | | | | |
| 0.5 | 0.5 | 91% | 77% | 0.70 | 0.68 |
| 0.3 | 0.7 | 90% | 79% | 0.72 | 0.67 |
| 0.1 | 0.9 | 85% | 62% | 0.77 | 0.73 |
| 0.05 | 0.95 | 76% | 60% | 0.81 | 0.76 |
| 0.01 | 0.99 | 30% | 58% | 0.85 | 0.82 |

Table 3: Automatic evaluation of fluency, formality transfer, and content preservation for informal-to-formal style transfer models using a linear combination of two objectives ($-\log p(\mathbf{y}|\mathbf{x})$ and USIM($\mathbf{x}, \mathbf{y}$)) with different weights. Since USIM lies in $[0, 1]$, it gets ignored if its weight is low, however increasing its weight compromises the fluency.

**USIM** USIM named after UKPLab-Sentence-Transformers is defined as USIM($\mathbf{x}, \mathbf{y}$) = cosine($M(x), M(y)$). In other words, it is the cosine similarity between the representations of a model $M$. This model is parameterized by GPT2(345M) [51]. $M(x)$ is obtained by first feeding $\mathbf{x}$ to the model and then mean pooling all the output representations. This model originally presented in Reimers and Gurevych [55] is trained in a Siamese fashion on BERT [41] but is easily extensible to any LM architecture. We adapt it to GPT2 as follows:

- First, we fine-tune $M$ =GPT2 on the combination of SNLI and MNLI [69] corpora which are both designed for training natural language inference model and intended to capture semantics. Each corpus contains pairs of sentencse with one of the three annotations: inference, contradiction or neutral. For each input sentence $(\mathbf{s}_1, \mathbf{s}_2)$, the model is trained as with classification objective with the final logits computed as $W[M(\mathbf{s}_1), M(\mathbf{s}_2), |M(\mathbf{s}_1) - M(\mathbf{s}_2)|]$, where $W$ is a trainable parameter. In other words the three vectors as shown are concatenated and multiplied with a weight matrix. We train this for 1 epoch on the combined corpora.

- Second, we continue fine-tuning the $M$ trained so far on the STS corpus which consists of pairs of sentences annotated with real numbers in $[-1, 1]$ indicating their semantic similarity. We train on this corpus with a mean-square-error loss between cosine($M(\mathbf{s}_1), M(\mathbf{s}_2)$) and the given score.

For details of training $M$ can be found in [55] where this model is shown to perform competitively on STS benchmarks [69]. We use this model for adding constraints in style-transfer (§3.1) and multi-attribute transfer (§4).

**XSIM** Similar to USIM, we define XSIM($\mathbf{x}, \mathbf{y}$) = cosine($CM(x), CM(y)$), where $CM$ is a cross-lingual model. This method was introduced by Reimers and Gurevych [56] where they distill a monolingual model such as $M$, to train a cross-lingual model with a small parallel corpus in the languages of interest. Given a parallel sentence pair $(\mathbf{x}, \mathbf{y})$, $CM$ is trained by minimizing the following loss:

$$\mathcal{L}_{\text{XSIM}} = \|M(\mathbf{x}) - CM(\mathbf{x})\|_2^2 + \|CM(\mathbf{x}) - CM(\mathbf{y})\|_2^2$$

That is, representations of the model $M$ and $CM$ for the source sentence are trained to be close together as are the cross-lingual representations of source and target. We parameterize $CM$ also with pretrained GPT2 (345M) [51] model. But GPT2 and the Marian Transformer based MT model [24] we use do not have matching vocabularies. Since the vocabulary of the primary objective and constraints should match for the decoding to work, we replace input word embedding layer of GPT2 with that of the decoder of the translation model before we train the distilled model. We use the TED2020 [] French-English parallel corpus containing around 400K sentence-pairs to train XSIM and obtain comparable performance as Reimers and Gurevych [56] on the cross-lingual STS benchmark [69].

**WMD** Given two bags of words, $x = \{x_1, \ldots, x_n\}$ and $y = \{y_1, \ldots, y_m\}$, and an embedding table $\mathbf{e}$, we define word mover's distance between $\mathbf{x}$ and $\mathbf{y}$ as

$$\text{WMD}(\mathbf{x}, \mathbf{y}) = \min \sum_{i=1, j=1}^{m,n} T_{ij} d_{ij} \text{ subject to}$$

$$\sum_i^n T_{ij} = \frac{1}{m}$$

$$\sum_j^m T_{ij} = \frac{1}{n}$$

where we define $d_{ij} = 1 - \cos(\mathbf{e}(x_i), \mathbf{e}(y_j))$. Given fixed inputs $\mathbf{e}(x_i)$ and $\mathbf{e}(y_j)$, WMD can easily be computed using linear program solver [9]. To backpropagate through this objective. We use the following steps following Kumar et al. [31]:

1. During the forward pass, we obtain $\hat{\mathbf{y}}$ as indicated in algorithm 1 and compute word embeddings for both the input $\mathbf{x}$ and the prediction $\hat{\mathbf{y}}$. Using the linear program solver, we compute $\text{WMD}(\mathbf{x}, \hat{\mathbf{y}})$ as well the proportions $T_{ij}$

2. During the backward pass, we keep the $T_{ij}$ fixed which removes the constraints from the WMD computation as described making it differentiable allowing gradients to flow to update the optimization parameters $\tilde{y}$.

We use the embedding table from USIM model as $\mathbf{e}$ for this constraint.

### D.2 Models used in multi-attribute transfer

In §4, we present a paraphrasing model with 4 different constraints: USIM as described previously and three classifier constraints. All the classifiers are trained by finetuning GPT2[10] on the following corpora:

**Age** We use the NUFA corpus [23] consisting Yelp Restaurant Reviews with 300K sentences per age group (greater than 30 years, and less than 30 years) in the training set. Our classifier achieves an accuracy of $\sim 80\%$ on a balanced test set of 10K sentences.

**Formality** We use GYAFC corpus as described in §3.1 for this constraint (with an accuracy of around 92%) on the provided test set.

**Sentiment** We collect Yelp restaurant reviews using scripts provided by Lample et al. [33][11] with a rating from 1 to 5 star. We subsample from this corpus to train our 5-class classifier on 100K reviews per rating obtaining a classification accuracy of around $75\%$ on a held-out test set also sampled from the same corpus.

## E More Details of Human Evaluation

We conduct A/B testing to rank translations generated by our method and beam search. We show the annotators the source sentence and two randomized translations (one from beam search and one from our method). We ask them to choose one of the four options: **1**: the first translation is both faithful and formal while the second is not, **2**: the second translation is both faithful and formal while the second is not, **3**: both are faithful and formal, and **4**: both are either unfaithful or informal or both. Results are summarized in §3.2.

---

[9] We solve it using the python library POT: `https://pythonot.github.io/`

[10] we use Huggingface [70] with recommended hyperparameters for training all classifiers: `https://huggingface.co/transformers/v2.0.0/examples.html`

[11] `https://github.com/facebookresearch/MultipleAttributeTextRewriting/tree/master/data/Yelp`

# F Examples

## F.1 Style Transfer

We show selected examples from our style-transfer models in Table 4. Since the final output $\mathbf{y}$ is generated from the paraphrase $\mathbf{z}$, not the input sentence $\mathbf{x}$, some of the content is at times modified in the final output in decoding without constraints. MUCOCO with content based constraints is able to recover content in some examples and also improve formality of the outputs. But it can still be prone to errors since the content-similarity metrics are not perfect. See §3.1 for more details.

| | |
|---|---|
| Input | all are good movies u cant rate them |
| Paraphrase | all good films you can't rate |
| NO-CONSTRAINTS | All good movies that you can't rate |
| FUDGE(FORMAL) | All good movies you can not rate |
| MUCOCO (FORMAL,USIM,WMD) | All are good movies you can not rate them adequately |
| Input | One guy says to me 'So, where's the fox from Mystic Pizza? |
| Paraphrase | "so, where's the Mystic Pizza guy?" Said one of the men. |
| NO-CONSTRAINTS | One of the men replied , "So , where is the Mystic Pizza guy? |
| FUDGE(FORMAL) | One of the men replied, "Where is the Mystic burger?" |
| MUCOCO (FORMAL,USIM,WMD) | One of the men enquired, "Where is the Mystic Pizza man?" |
| Input | chill out sweetie everything will be fine eventually. |
| Paraphrase | calm down, sweetheart, everything will be fine. |
| NO-CONSTRAINTS | Calm down, sweetheart, everything is going to be alright. |
| FUDGE(FORMAL) | Calm down sweetheart, everything will be alright. |
| MUCOCO (FORMAL,USIM,WMD) | Calm down sweetheart, everything will eventually proceed to be good. |
| Input | what kinda person are you to even ask that ridiculous (er...spell check that...) question? |
| Paraphrase | what kind of fool would ask such a silly question? |
| NO-CONSTRAINTS | What kind of idiot would ask such a dumb question? |
| FUDGE(FORMAL) | What kind of fool would ask such a dull question? |
| MUCOCO (FORMAL,USIM,WMD) | What kind of foolish person would ask such a dull question? |
| Input | I spent my money, enjoyed myself and didn't loose more than I could afford - I won.... |
| Paraphrase | I've spent my money, I've enjoyed it, and I didn't lose anything. |
| NO-CONSTRAINTS | I spent my money, I enjoyed it, and I did not lose anything. |
| FUDGE(FORMAL) | I have spent my money, I have enjoyed it, and I did not lose anything. |
| MUCOCO (FORMAL,USIM,WMD) | I spent my money, did not lose anything more, and it was simply enjoyable. |

Table 4: Style transfer examples with different decoding methods and constraints.

## F.2 Style-controlled Machine Translation

Table 5 lists few selected examples for inducing cross-lingual similarity and formality constraints in a French to English MT model. We find that inducing formality modifies some of the constructs (like removing contractions: "gonna" to "going to") in the output sentences which are not measured accurately by a surface-level metric like BLEU. See §3.2 for more details.

## F.3 Multiple Solutions on the Pareto Front

Table 8 shows a few examples of changing constraint thresholds for semantic similarity as well as formality constraints. Since the classifiers are not well calibrated, we find that with tighter constraints, the outputs tend to overly represent formality indicating words while losing some of the content which the semantic similarity models are not always robust enough to detect. See §4 for more details.

| | |
|---|---|
| Source | Mais il s'agit... il s'agit d'une femme que vous ne connaissez pas. |
| Reference | But this is– This is a woman you don't know. |
| BEAMSEARCH | But this is... this is a woman you don't know. |
| MUCOCO (XSIM) | But this is... this is a woman you don't know. |
| FUDGE(FORMAL) | But this is... this is a woman you do not know. |
| MUCOCO (FORMAL) | But this is... is a woman you do not know. |
| MUCOCO (FORMAL,XSIM) | But this is a woman you do not know. |
| Source | Toi ? Le mec à bananes, exact. |
| Reference | - Who's the banana man, alright. |
| BEAMSEARCH | You, the banana guy, right. |
| MUCOCO (XSIM) | You? the banana guy, right. |
| FUDGE(FORMAL) | You, the banana guy, right? |
| MUCOCO (FORMAL) | Are you the banana guy? |
| MUCOCO (FORMAL,XSIM) | Are you the banana guy? |
| Source | Nous allons les sortir de la d'ici quelques minutes. |
| Reference | We'll have them out in a couple minutes. |
| BEAMSEARCH | We're gonna get them out of here in a few minutes. |
| MUCOCO (XSIM) | We're gonna get them out of here in a few minutes. |
| FUDGE(FORMAL) | We'll get them out of here in a few minutes. |
| MUCOCO (FORMAL) | We will get them out of here. |
| MUCOCO (FORMAL,XSIM) | We will get them out of here in a few minutes. |
| Source | On va prendre la voie aérienne. |
| Reference | We'll take the aerial up. |
| BEAMSEARCH | We're gonna take the airway. |
| MUCOCO (XSIM) | We're gonna take the air route. |
| FUDGE(FORMAL) | We are gonna take the airway. |
| MUCOCO (FORMAL) | We are going to take the air. |
| MUCOCO (FORMAL,XSIM) | We are going take the air route. |
| Source | Mais mon sang ne correspondait pas. |
| Reference | But my blood didn't match. |
| BEAMSEARCH | But my blood wasn't matching. |
| MUCOCO (XSIM) | But my blood didn't match. |
| FUDGE(FORMAL) | But my blood wasn't matched. |
| MUCOCO (FORMAL) | But my blood was not correct. |
| MUCOCO (FORMAL,XSIM) | But my blood did not match. |

Table 5: Translation examples with different decoding methods and constraints.

## F.4 Multi-attribute Transfer

Table 6 shows a few examples of transfering multiple combinations of attributes in a given input sentence. We focus on sentiment amplification rather than transfer as it is by definition prone to losing content (See table 7 for an example). See more details in §4.

| < 30 years, informal, 4-star | one big plus : the coffee is always fantastic . |
|---|---|
| < 30 years, informal, 5-star | the coffee is always great ! |
| < 30 years, formal, 4-star | this coffee is incredibly good. |
| < 30 years, formal, 5-star | the coffee is consistently outstanding! |
| > 30 years, informal, 4-star | the espresso is usually enjoyed . |
| > 30 years, informal, 5-star | the coffee is usually delicious also! |
| > 30 years, formal, 4-star | the espresso is pleasantly delicious, nonetheless. |
| > 30 years, formal, 5-star | the coffee is brewed to excellence. |
| **< 30 years, informal, 2-star** | **i left our meal feeling a little disappointed .** |
| < 30 years, informal, 1-star | worst feeling with this little meal . |
| < 30 years, formal, 2-star | i felt failed and disappointed by this meal . |
| < 30 years, formal, 1-star | i left our meal feeling anguished, betrayed . |
| > 30 years, informal, 2-star | i was a little disappointed ! |
| > 30 years, informal, 1-star | this meal bummed me out ! |
| > 30 years, formal, 2-star | i felt unsatisfied by this meal. |
| > 30 years, formal, 1-star | i felt complete disappointment after this meal . |

Table 6: MUCOCO with multiple constraints and rewriting reviews with different combination of attributes.

| < 30 years, informal, 2-star | i left our meal feeling a little disappointed . |
|---|---|
| < 30 years, informal, 5-star | i was excited when I left |
| < 30 years, formal, 5-star | i was impeccably good |
| > 30 years, informal, 5star | i was extremely amazing. |
| > 30 years, formal, 5-star | i was exquisite and a bit phenomenal |

Table 7: MUCOCO with sentiment transfer instead of amplification. We remove the USIM constraint here as it gets violated. Without that constraint, we observe that while sentiment transfer is achievable, it substantially alters the meaning of the input text.

| Input Sentence | My dad looks like Paul Newman, and my ex looked like king kong |
|---|---|
| Paraphrase | my dad's like Paul Newman, and my ex looks like a king. |

| **Constraints** | **Outputs** |
|---|---|
| FORMAL($\mathbf{y}$) > 0.5, USIM($\mathbf{x}, \mathbf{y}$) < 0.15 | My dad looks like Paul Newman, and my ex looks similar to King Kong |
| FORMAL($\mathbf{y}$) > 0.7, USIM($\mathbf{x}, \mathbf{y}$) < 0.15 | My father looks like Paul Newman, and my ex resembles a King Kong |
| FORMAL($\mathbf{y}$) > 0.9, USIM($\mathbf{x}, \mathbf{y}$) < 0.15 | My father looks like Paul Newman, and my ex possesses the qualities of King Kong approximately |
| FORMAL($\mathbf{y}$) > 0.7, USIM($\mathbf{x}, \mathbf{y}$) < 0.1 | My dad possesses looks similar to Paul Newman, my ex appears like King King Kong |
| FORMAL($\mathbf{y}$) > 0.9, USIM($\mathbf{x}, \mathbf{y}$) < 0.05 | My dad possesses the Paul Newman looks similar my ex possesses similar King Kong resemblance |

Table 8: Varying thresholds for the constraints to find other solutions on the Pareto front.