# OpenReview forum: "Controlled Text Generation as Continuous Optimization with Multiple Constraints"
_NeurIPS.cc/2021/Conference — NeurIPS 2021 Poster_

### Official Review · Reviewer_deJ9 · 2021-07-14

**Rating:** 7
**Confidence:** 4

**Summary:**

In this paper, the authors propose a controllable decoding mechanism for language generation models (e.g., GPT2) using constrained optimization. The proposed method explicitly formulate the sentence decoding as a constraint optimization problem where the free parameters are the distribution over the vocabulary at each step and two types of constraints are defined one over the generate text itself (f(Y)) and one over the similarity between the input and the outputs (g(x,y)). Please refer to Eq. 1 where the problem is clearly formalized. Since directly solving the problem in Eq 1. is not computationally feasible the author reformulates the problem using the Lagrangian formulation for container optimization problems (Eq. 3). However, this min-max formulation introduces several problems (e.g., the tradeoff between different objectives, hard to find opt for non-convex functions). To cope with these issues, the authors proposed a gradient descent/ascent solution (Section 2.1), a soft relaxation for each token using the probability simplex (exponential gradient descent + normalization) (Section 2.2), and an annealing threshold (Section 2.3).

The authors benchmarked the proposed decoding schema in both style transfer and style-controlled machine translation. In the style transfer, they follow the methodology in Krishna et al. 2020 and they study two contains such as formality and the semantic similarity between the input and the output. Similarly in the style-controlled machine translation, they study both the formality style and the cross-lingual similarity between input-output. In both settings, the author compared the proposed method with other decoding techniques (greedy and beam-search) and other controllable language generation techniques (e.g., FUDGE) using both automatic and human evaluation.  Finally, the authors show some preliminary results in controlling multiple attributes (Age, style pos/neg), and discuss the speed and memory requirements of the model.

**Ethical Concerns:**

The authors addressed this in the ethical considerations section.

**Limitations And Societal Impact:**

The author clearly states the limitation, i.e., a very slow inference time, and possible workaround in future work. Moreover, they also address possible negative societal impacts in the ethical considerations section.

**Main Review:**

***Originality***

To the best of my knowledge, the proposed method is novel in controllable language generation. Moreover, constrained optimization is very interesting in this setting and this is very different from existing approaches. However, I would suggest including some relevant work as [1,2]

***Quality***

The paper is technically sound, but I would have expected more experiments and comparisons with existing work. Indeed, this method is general and it could be used with any discriminator (e.g., positive/negative, topics, toxicity etc.). Several existing works (e.g. GEDI, PPLM etc.) run these controllable language generation experiments. Adding these experiments would have made the paper much stronger and more general. The results on formalization are not very exciting, although the result is extremely good. Moreover, I am a bit suspicious when checking Appendix Table 5,  where the authors show the multiple constraints example, in that example, I would be curious to see what happen when rewriting a review with an original score of 2 to a review with a score of 5. Is this possible?

Moreover, it is also not clear the human evaluation for the style-controlled machine translation (Line 295). Why not running the same human eval as in the style transfer experiments? and the results in percentage are really confusing. Could you please clarify the human evaluation.

***Clarity***
The paper is extremely well written,  the notation is clear and easy to follow also with a minimal background in optimization.


***Significance***:
Yes, this proposed method is extremely important for any general NLG model. In general, enforcing constraints in text generation is extremely hard and solving this problem could be crucial for safely deploying large language models.


[1] NEUROLOGIC DECODING: (Un)supervised Neural Text Generation with Predicate Logic Constraints (Lu et. al. 2020)
[2] Gradient-guided Unsupervised Lexically Constrained Text Generation (Sha et.al. 2020)


**Time Spent Reviewing:**

4

---

> ### Author Response · Authors · 2021-08-10
> **Response to Reviewer deJ9**
>
> We thank the reviewer for their extremely valuable feedback. Here are our responses to your questions/comments
>
> Comparison to more existing works and tasks: We thank you for providing the references [1,2] and will make sure to cite and discuss them in the updated draft. However, these are not suitable  baselines to our approach as they are limited to lexical controls in the generated text, rather than underlying text attributes as we focus on this work. On the other hand, experiments related to controls like topic and toxicity (as explored in PPLM, GEDI) are open-ended generation tasks relying on sampling from the language models like GPT2 given a prompt with some additional constraints. In our current approach however, we try to maximize the likelihood of the generated text according to the language model. Sampling from the LM while satisfying the constraints via gradient descent is not trivial and we plan to explore this direction in our future work.
>
> Flipping sentiment: In its formulation, it is possible to rewrite an input to change the sentiment from a score of 2 to 5, however it will naturally flip the meaning (as measured by the semantic similarity models) and the similarity constraint will likely fail to satisfy. Removing this constraint can in fact generate sentences with the flipped sentiment. Here is an example:
>
> Input sentence: i left our meal feeling a little disappointed  (<30 years, informal, 2-star)
>
> Target constraints: age=<30 years, formality=informal, sentiment=5-star
>
> Predicted output: i was excited when I left
>
> Target constraints: age=<30 years, formality=formal, sentiment=5-star
>
> Predicted output: i was impeccably good
>
> Target constraints: age=>30 years, formality=informal, sentiment=5-star
>
> Predicted output: i was extremely amazing.
>
> Target constraints: age=>30 years, formality=formal, sentiment=5-star
>
> Predicted output: i was exquisite and a bit phenomenal
>
> In this example, the meaning is substantially changed from the narrator talking about the meal to just talking about themselves.
>
>
> Details of human evaluation: In the translation experiments, we evaluated using a setup similar to A/B testing. Given predicted translations from two inference methods (ours vs a baseline), human judges were asked to look at both the source sentence and the predicted translations (which were randomized) and rate them based on two criteria: faithfulness and formality. That is, they had to choose amongst the following options: (1) Both translations satisfy both criteria (2) Only one of the translations satisfies both criteria or (3) Neither solutions satisfy both criteria. We do not evaluate on the two criteria independently since the prediction being formal does not matter if it is not faithful and hallucinates content. The numbers reported in the paper indicate the percentage of the annotations corresponding to each of the options mentioned above. Due to space limitations, we have included these details in the Appendix.
>
> [1] NEUROLOGIC DECODING: (Un)supervised Neural Text Generation with Predicate Logic Constraints (Lu et. al. 2020)
> [2] Gradient-guided Unsupervised Lexically Constrained Text Generation (Sha et.al. 2020)

---

> > ### Comment · Reviewer_deJ9 · 2021-08-18
> > **Re: Review**
> >
> > Thanks for your clarification. Especially could you please mention in the new version of the paper:
> > 1- the examples of flipping the sentiments
> > 2- limation of your work "Sampling from the LM while satisfying the constraints via gradient descent is not trivial and we plan to explore this direction in our future work.", this will help future researchers to build and improve your method.

---

> > > ### Author Response · Authors · 2021-08-20
> > > **Re: Re: Review**
> > >
> > > Thank you for your reply. We will definitely update the final draft with these details.

---

### Official Review · Reviewer_CUjF · 2021-07-15

**Rating:** 7
**Confidence:** 3

**Summary:**

This work tries to solve a general type of text generation tasks, generating text with constrains. The author proposed a a new algorithm MUCOCO to formulate the decoding process as a constrained optimization problem. The author then use continuous relaxation, Lagrangian multipliers and exponential gradient descend techniques to solve the problem. The author validated the algorithm with style transfer and style-controlled machine translation.

**Main Review:**

This paper formulates the constrained text generation problem into a discrete constrained optimization task. This idea is quite novel and interesting. The proposed setup of Lagrangian multiplier, straight through and exponential gradient descent are natural tools to use for this setup.

I do not get the reason why the author emphasis the proposed framework is for pretrained model. From the method itself, I do not see a limitation on this.

As the proposed algorithm is to improve over auto-regressive model which is hard to follow the constrains. Then the other type of framework,  non-autoregressive text generation should be interesting to be considered in this work, at least as a baseline for MT task.

I do not see details in inference stage, will the proposed algorithm effect decoding stage?

The method's effectiveness seems strongly relay on the auxiliary optimization objective f and g. This may leads to a limitation to the proposed method, how to define a differentiable objective can be a hard task. Also in the experiments, seems the function threshold needs carefully selection, which may make the proposed algorithm practically hard to use. An empirical study on these should be shown in the paper.

**Time Spent Reviewing:**

5

---

> ### Author Response · Authors · 2021-08-10
> **Response to Reviewer CUjF**
>
> We thank the reviewer for their valuable feedback. Here are our responses to your comments and questions.
>
> The proposed method in this work is indeed an inference method to decode from already existing language models such that certain constraints are satisfied, and not train a new generation model. We consider fine-tuned versions of large LMs like GPT2 in this work to ensure fluency and coherence, in addition to satisfying desired constraints. However, that is not a requirement; any language generation model that’s already trained can be used with this approach. Hence, the emphasis on pretrained models.
>
> While non-autoregressive MT also focuses on generating the entire output sequence at once, their goal is to improve the decoding speed and to the best of our knowledge, it is not trivial to extend it to include constraints. On the other hand, our proposed method is slower but is useful to enforce desired constraints.
>
> Differentiable constraint functions and thresholds: While we agree that it is a strict requirement, most examples of sentence level constraints considered in prior work on controllable text generation are indeed differentiable ([1, 2, 3] among others) and can be easily applied in our method. The thresholds for the constraints although being hyperparameters are not hard to tune. As we discussed in section 3, for classifier based constraints like formality, we simply set it to be >0.5 for the desired class. And for semantic similarity based constraints, we use their average value on a development set as the threshold and it seems to work well. We also provide a discussion on how changing the values of the thresholds changes the final outputs in section 4 (titled “other solutions on the Pareto front”). The straightforward selection of constraints is in fact, an advantage of the proposed approach over the alternative of using a linear combination of constraints which would, indeed, require automatic hyperparameter tuning.
>
> [1] PPLM: Plug and Play Language Models: A Simple Approach to Controlled Text Generation. Dathatri et al 2019.
>
> [2] FUDGE: Controlled Text Generation With Future Discriminators. Yang et al 2021.
>
> [3] DExperts: Decoding-Time Controlled Text Generation with Experts and Anti-Experts. Liu et al 2021.

---

> > ### Comment · Reviewer_CUjF · 2021-09-06
> > **Re: Response**
> >
> > Thanks for the response. My concerns have been mostly resolved.

---

### Official Review · Reviewer_DZSf · 2021-07-16

**Rating:** 6
**Confidence:** 3

**Summary:**

This paper proposes a flexible controllable decoding algorithm called MUCOCO, which incorporates multiple control attributes as differentiable constraints to the optimization. The authors relax the discrete optimization problem to a continuous one, and use continuous optimization techniques such as Lagrangian multipliers and gradient-descent based techniques to generate texts towards the desired attribute. Experiments on three conditional text generation tasks including text style transfer, machine translation and paraphrasing show the superior performance of the proposed model.

**Limitations And Societal Impact:**

The authors adequately address the limitations and potential negative societal impact of their work.

**Main Review:**

Strengths:
1) The formulation of constrained decoding is intuitive and interesting. It provides a new perspective to solve the text generation problem under multiple constraints.
2) Experiments on three different text generation tasks show the generalization ability of the proposed method.

Weaknesses:
1) My main concern is about the focus of this paper. I understand that the formulation itself is interesting. But the following techniques such as continuous relaxation and gradient-based optimization which are the main content of the method are all common techniques in the machine learning community, which don’t provide new insights. So, I feel that the focus should be addressed clearly.
2) The authors choose FUDGE as the main baseline, but this model can only support a single constraint. The authors should consider the baselines which can handle text generation with multiple control attributes such as CoCon [1].
3) In my view, one of the major problems in text generation with multiple control attributes is the compatibility of multiple constraints. I wonder how the proposed model can explicitly deal with this problem.

[1] COCON: A SELF-SUPERVISED APPROACH FOR CONTROLLED TEXT GENERATION. ICLR 2021.

**Time Spent Reviewing:**

4

---

> ### Author Response · Authors · 2021-08-10
> **Response to Reviewer DZSf**
>
> We thank the reviewer for their valuable feedback. Here are our responses to your comments and questions:
>
> Focus of the paper: We agree with your assessment that the techniques we employ in our presented approach (including the Lagragian, continuous relaxation and gradient-based optimization) are well-known in the ML community. However, our contributions are not the techniques themselves but rather their novel combination and application to the task of controllable text generation. This task has been explored in many recent works with several heuristic solutions proposed (including linear combination of objectives). In this work, we propose a more principled and general approach which is grounded in literature. We thank you for pointing this out, we will describe it more clearly in the final draft.
>
> Baselines: Our main baseline FUDGE can in fact support multiple constraints as reported by the authors in one of their experiments (couplet generation). However, their setup has two shortcomings: (a) constraints are enforced only to the text generated so far (as opposed to the entire output) since they generate left to right and, (b) the constraints should be representable via probabilities (like classifiers). We try to address both of them in this work as we generate the entire output together and allow for any real-valued functions as constraints. Thank you for pointing us to the cited paper (CoCon), we were not aware of it at the time of submission and will make sure to discuss it in the final draft. However, we believe this paper does not serve as a baseline for our work due to the following reasons: (a) The authors’ goal in the paper is NOT to decode with constraints from already trained models (like we do), but to train language generation models to follow specific constraints similar to work like CTRL [1]. (b) The kind of constraints they deal with in the paper are limited to word/phrase level where the generated output is expected to contain specific tokens as indicated by a prefix. For example, to generate a text with negative sentiment, they provide “is horrible” as a prefix to the model. This formulation is limiting and is not amenable to be adapted to the constraints we use.
>
> Compatibility of the constraints: In our current setup, incompatible constraints where satisfying one will violate the other, will likely lead to the optimization not being successful (the Lagrange multipliers will take extreme values). When the optimization fails, we just output the input sentence (for style transfer) or the unconstrained translation (for machine translation), instead of predicting a nonsensical output. Nonetheless, we argue that choosing an appropriate set of constraints should be a design choice made by the practitioner. We will include this discussion in the final draft as well.
>
> [1] CTRL: A Conditional Transformer Language Model for Controllable Generation. Keskar et al 2019

---

> > ### Comment · Reviewer_DZSf · 2021-08-21
> > **Reply to the author response**
> >
> > Thanks for your response. My concerns have been almost solved. I hope that the authors can clearly address the focus of this paper, and add more discussion about baseline selection and compatibility of multiple constraints in the final version. I will increase my rating.

---

> > > ### Author Response · Authors · 2021-08-23
> > > **Thank you**
> > >
> > > We thank you for your response and appreciate your feedback. We will make sure to clearly address the mentioned issues in the final version.

---

### Official Review · Reviewer_LkvM · 2021-07-23

**Rating:** 6
**Confidence:** 4

**Summary:**

This paper presents a decoding algorithm for controlled text generation. The problem is formulated as finding the most likely output $y$ under given constraints. The authors adopt the method of Lagrange multipliers, and introduce dampening in gradient descent to reduce oscillations. To deal with discrete $y$, the authors choose to optimize over probability simplex $\tilde y$ using exponentiated gradient descent. The authors also report that converting $\tilde y$ to a one-hot vector via a straight through estimator helps to converge faster.

Experiments include increasing formality in style transfer and machine translation. Compared with greedy decoding/beam search, the proposed method with formality and semantic similarity constraints achieves better content preservation/BLEU score and better transfer accuracy. It also compares favorably to the recent FUDGE method for controlled generation.

**Limitations And Societal Impact:**

Limitations and societal impact are discussed in Sec 4.

**Main Review:**

Constrained text generation is an important problem that has received increasing attention. This paper proposes a multi-objective optimization approach for constrained decoding, and motivates the choice of each technique used.

Is the formality classifier used in optimization also used for evaluation? If so, as you set the probability threshold to 0.5, if the constrained optimization was successful, shouldn't you achieve formality 100%? Why in Table 2 it's only 30%? Or the optimization failed and ended up with Lagrangian multipliers of extreme values?

In the method section, the authors discussed the shortcomings of using linear combinations of multiple objectives and advocated their Lagrangian approach. Nevertheless, I still would like to see how linear combinations perform in the experiments. Also, if you apply your continuous optimization in the unconstrained case (so maximizing likelihood is the only objective), how would it compare to beam search? I know it's slower than beam search, but will its BLEU be better or worse? I think this result will help to better understand the comparison between continuous optimization and combinatorial search.

Typo: line 127, "$(\mu_j-g_j(x,y))$" should be $(\xi_j-g_j(x,y))$

**Time Spent Reviewing:**

5h

---

> ### Author Response · Authors · 2021-08-10
> **Response to Reviewer LkvM**
>
> Thank you so much for your feedback. Please find our responses to your questions and comments below:
>
> Formality Evaluation: The formality classifier used for evaluation (fine-tuned RoBERTa-Large)  is NOT the same as the one used during optimization (fine-tuned RoBERTA-base) which can result in some drop in performance. But as you have correctly pointed out, this drop is largely due to optimization not always leading to the constraints getting satisfied and the Lagrangian multipliers taking extreme values. We hypothesize this is because of the highly non-convex nature of the optimization objective which might sometimes lead to the gradient descent getting stuck in a local minimum.
>
> Linear combination: With a motivating example shown in Figure 1 in the paper, we discussed that a linear combination can completely ignore some of the objectives especially if they lie on different scales. In our experiments, we observe this behavior both with xsim and usim as constraints (since they lie in [0, 1]) whereas the LM objectives are in [0,inf]. As a result, the outputs with this approach had almost exactly the same scores if we didn’t use any additional objective at all. Hence, we did not include this result in the draft. We will discuss this in more detail in the final draft. As an example, we obtain the following results when using the usim constraint in style transfer experiments with different combinations of weights:
>
> Weights (LM:usim)	  | 0.5:0.5   | 0.3:0.7     | 0.1:0.9 | 0.05:0.95 | 0.01:0.99
>
> Fluency (%)		  |   91%	  | 90%	    | 85%     | 76%	   | 30%
>
> Transfer (%)		  |   77%	  | 79%	    | 62%     | 60%	   | 58%
>
> wsim (wrt reference) |   0.68	  | 0.67	    | 0.73     | 0.76	   | 0.82
>
> wsim (wrt source)	  |   0.70	  | 0.72         | 0.77     | 0.81	   | 0.85
>
> For comparison, here are the results we reported in the paper:
>
> Metrics | No constraints | MuCOCO with usim
>
> Fluency (%)		  | 	91%  	    |      92%
>
> Transfer (%)		  |	78%	            |      85%
>
> wsim (wrt reference) |	0.69  	    |      0.71
>
> wsim (wrt source)	  | 	0.72  	    |      0.77
>
> As shown in the results, we observe that whatever weight we assign to usim, it either gets ignored if the value is not large enough, or overtakes the optimization and leads to solutions where the other objectives suffer (as observed by a sharp decline in fluency and transfer accuracies).
>
> Unconstrained Case: This has already been studied in prior work [1] and has been shown to perform marginally better than beam-search. We also had similar findings in our preliminary experiments. We chose not to include this result as our focus is on adding constraints. We will elaborate on this in the related work section.
>
> [1] Towards Decoding as Continuous Optimisation in Neural Machine Translation. Hoang et al 2017.

---

### Decision · Program_Chairs · 2021-09-27

**Decision:**

Accept (Poster)

**Comment:**

This paper proposes MUCOCO, a constrained text generation method. The constrained text generation is formulated as generating text with multiple controlling attributes. MUCOCO turns a discrete constrained decoding into differentiable optimization using techniques of  continuous relaxation, Lagrangian multipliers and exponential gradient descend. Experiments on three conditional text generation tasks including text style transfer, machine translation and paraphrasing show the superior performance of the proposed MUCOCO.

The author may adjust the main focus a bit accordingly. Additional discussion and/or comparison with other highly related methods (methods that can also handle constrained sentence generation with multiple constraints) could be added as pointed out by reviewers. The authors may explain the inference clearly. Additional discussion on limitations of MUCOCO could be added (e.g. conditions for auxiliary objectives and compatibility of multiple constraints) .